# Identifying Differences in Consumer Attitudes towards Local Foods in Organic and National Voluntary Quality Certification Schemes

**DOI:** 10.3390/foods12061132

**Published:** 2023-03-08

**Authors:** Hristo Hristov, Karmen Erjavec, Igor Pravst, Luka Juvančič, Aleš Kuhar

**Affiliations:** 1Nutrition Institute, SI-1000 Ljubljana, Slovenia; 2Faculty of Business and Informatics, University of Novo Mesto, SI-8000 Novo Mesto, Slovenia; 3Biotechnical Faculty, University of Ljubljana, SI-1000 Ljubljana, Slovenia

**Keywords:** quality schemes, Selected Quality, organic label, trust, segmentation, local food, consumer attitudes

## Abstract

As no study on attitudes towards local food has compared “organic” and national quality scheme consumer segments, this study aimed to provide further insights and clarifications on the issue of consumer segmentation in terms of trust towards organic food and food of selected quality perceived as local, along socioeconomic characteristics, and other important determinants of this complex interaction. The research examines consumers’ attitudes and perceptions related to two quality schemes for special Slovenian foods: “Organic”, which relates to production methods; and “Selected Quality”, which relates to quality attributes. The study focused on two segments of consumers, who exhibit a high level of trust towards the two quality schemes. Comparative analysis of the consumer segments looked for the potential differences with respect to their sociodemographic profiles, as well as to their understanding of the definition of local food, attitudes towards local food, trust in actors and institutions, and willingness to purchase local food. The study combined qualitative approaches and a quantitative survey with a general population sample. The results showed that both consumer groups have similar understandings of local food, with region-based interpretations outperforming country-based interpretations. The “Organic” group was more cosmopolitan and supportive of the local community, regardless of geographic proximity, than the “Selected Quality” group. Older consumers occupy a larger share of both segments, with professionals and individuals with higher incomes more likely to be in the “Organic” group and retirees and students more likely to be in the “Selected Quality” group. To increase the consumers’ interest in food with the “Organic” and “Selected Quality” schemes, more specific product propositions should be developed.

## 1. Introduction

Food quality certifications are gaining importance in shaping local and global agricultural and food systems [1]. Such schemes exist in a variety of institutional forms, have diverse origins, and may serve multiple purposes. The schemes differ in several attributes, including the extent to which users are free to decide and act to comply with the standards (e.g., mandatory vs. voluntary), and the role of public and private organizations in publishing and/or enforcing the standards. These very heterogeneous approaches of different food system actors (governments, farmers, food companies, retailers, industry associations, NGOs, etc.) can be grouped under the term Quality Assurance and Certification Systems (QACS). They are defined as a set of activities and/or facilities that enable food-supply-chain actors to ensure compliance with a specific standard, code of conduct or set of specified requirements, and to signal these attributes to the end consumer or a subsequent buyer in the supply chain.

Additionally, most EU member states and many regions, as well as private providers, are implementing national QACS in the above-mentioned category of voluntary certification schemes [2,3], also aiming to promote local producers. In 2014, Slovenia introduced the national voluntary quality scheme “Izbrana kakovost” (transl. “Selected Quality”), which is awarded to above-standard quality foods of local origin. This scheme is considered a cornerstone of agricultural products produced and processed in Slovenia and was successfully promoted among consumers. Given the complexity and dynamic nature of food quality certification and labeling, the effectiveness of promotional efforts depends heavily on reliable information, an understanding of the targeted consumers, and the food system. A better understanding of how consumers distinguish between origin and organic concepts is therefore also important for setting marketing strategies and evidence-based policy planning. 

There is growing evidence that food consumption preferences for organic food and food of local origin are related in the sense that consumers who value locally produced food have preferences for organic food [4,5,6], which translates into a positive willingness to pay a price premium for the attributes of local origin and organic food [7]. Additionally, interest in information about quality, traceability, sustainability, and origin also influences consumers’ purchase intentions of organic food [8,9,10]. All of this raises the question of whether organic and local origin are competing or independent attributes, when it comes to food purchasing [4,5,11,12,13]. Recent studies have not provided a clear answer to this question. In fact, some studies show that organic consumers are attracted to origin claims and consumers of local food are attracted to environmental claims [13,14], while other studies show that consumers prefer to buy local rather than organic food [5] or that consumers who perceive the benefits of organic products prefer both organic and local origin foods, while those who perceive the benefits of local origin do not necessarily prefer organic food [11]. There are different definitions of attitudes towards food of local origin—in consideration of cognitive, affective, and behavioral components. Among them, a ‘locavorism’ dominates as a preference for food of local origin [15]. A ‘locavore’ or committed consumer of local food believes that local food tastes better and is more nutritious (lionization); he/she is opposed to distant food systems (oppositions) and wants to support local communities (communalization) [15]. Many authors emphasize that consumers’ attitudes towards locavorism depend on how consumers interpret the term ‘local origin’ [15,16,17]. However, especially in a small country like Slovenia, which was the subject of this study, geographical proximity can be very ambiguous, and the crucial question is therefore how consumers understand food of local origin. 

In the context of reducing the information asymmetry, trust—defined as “a psychological state that involves the intention to accept vulnerability based on positive expectations of the intentions or behaviors of another” [18]—can be an important concept in driving loyalty and new solid relationships between producers and consumers [19]. Macready et al. [20] disaggregate consumer trust into (1) general trust (how trusting an individual is of others), (2) food chain trust (trust in food chain actors), (3) organizational trust (trust related to food organizations), and (4) product trust (trust in specific food products). Considering that trust is a key factor in consumer behavior and preference for a particular type of food, it is worth exploring this in relation to consumers’ opinions on organic food or food of local origin, which is measured and conceptualized differently across disciplines [21]. 

Having these challenges in mind, this study intends to contribute to further knowledge about the awareness, understanding, discrimination, perception, and evaluation of two categories of QACS: namely, the “European/national scheme for organic farming” (“Organic” scheme) and the voluntary national scheme “Selected Quality”. The study includes several research questions related to the existence of differences in the sociodemographic profile among the two segments of consumers, namely: (1) “trust in the Organic food label group” and (2) “trust in the national quality scheme Selected Quality”, and differences in the understanding of the definition of local food, attitudes towards local food, trust in actors and institutions/companies, and willingness to buy local food. Consumers’ attitudes towards local food, and their trust and purchasing habits will also be investigated in relation to trust towards the “Organic” and “Selected Quality” schemes and compared against the behavior of the general population sample. 

This paper thus aims to provide further insights and clarifications on the issue of consumer segmentation in terms of preferences for organic food or food of local origin, along socioeconomic characteristics, and other important determinants of this complex behavioral phenomenon. Undoubtedly, a comprehensive understanding of this phenomenon is essential for more effective planning of marketing strategies by business actors, and policy design and implementation by public authorities.

## 2. Materials and Methods

The study is based on qualitative and quantitative approaches to identify possible significant patterns of differences between the two consumer segments studied and compare them with the general, nationally representative population. In the first phase, focus groups were conducted with participants selected according to gender, age, education, income, and place of residence (rural/urban) using the snowball method. The purpose was to explore the relationships between consumer opinions and consumption motives. The focus groups included 8–10 persons of different sociodemographic backgrounds. They were conducted with prior consent of the participants in a video conferencing format (Microsoft Teams software) during the winter of 2020/2021, as COVID-19 preventative measures prevented us from conducting live focus groups. The focus group discussions, which lasted about 2 h on average, focus on three main topics: (1) factors influencing the consumption of certified “Organic” and “Selected Quality” labeled foods, (2) consumer preference for certified “Organic” and “Selected Quality” labeled foods, and (3) the ranking of these attributes to determine their relative importance. The results were further tested in a real-life situation with a subset of 16 participants through the method of shopping with consumers to identify direct experiences in the retail environment [22]. The actual shopping took about one hour, during which, we took photos, and recorded conversations and observations. The observation was, in particular, on their purchases of labeled food. Additionally, we asked participants to indicate their motivations for buying certain foods. In the second phase, a quantitative survey was conducted to investigate the extent to which consumers’ opinions about organic farming and the voluntary national quality scheme “Selected Quality” identified in the focus groups are reflected in the nationally representative sample of the Slovenian population. The quantitative survey aimed to identify sociodemographic patterns and differences or similarities in the understanding of the definition of local food, attitudes towards local food, trust in actors and institutions/businesses, and willingness to buy local food between consumer segments.

### 2.1. Quantitative Data Collection and Measuring Scales

The quantitative data collection was conducted using an on-line questionnaire. Subjects were recruited from a consumer panel of a marketing research institute, which consists of about 35,000 subjects from Slovenia. We used quota sampling for age groups, gender, and place of residence to ensure that the sample structure is comparable with the Slovenian population. The study sample included 1007 participants, aged between 18 and 65. All the data was collected in May and June 2021. Table 1 shows the sociodemographic characteristics of the study sample.

The study design of the quantitative survey included two interventions in the original sample, which was randomly divided into two equal samples that controlled for age, gender, and place of residence. Each participant of the two samples was asked to identify the five most important attributes associated with certified organic foods and food with the national quality scheme label “Selected Quality”. Additionally, the original sample was randomly divided into five subsamples, using quotas for age, gender, and place of residence. In each subsample, the respondents were asked to indicate the importance of various attributes when purchasing local foods in five different food categories. All respondents were asked to answer all survey questions, considering the areas of interest specified by the study design.

The draft questionnaire was based on existing academic literature on the topic, focus group discussions, and the results of shopping with consumers. It was pretested by six leading Slovenian food experts and checked for correct transcription and comprehension in a pilot study with over 100 participants.

The participants were asked to complete a self-administered structured online questionnaire with six blocks. The first block dealt with sociodemographic factors (age, gender, education, income, occupational status, place of residence (geographical region)). The second block focused on understanding the definition of local food, which was measured using scales adopted from previous studies [16,17,23]. Five items were included that focused on the geographic proximity of production in general (local/regional and national), the location of producers (local farmers and manufacturers), farm-to-table sales, and traditional food production or specialty, using a 7-point interval scale ranging from 1 (strongly disagree) to 7 (strongly agree). Furthermore, nine items of locavorism [15] were used including lionization (better tasting and more nutritious locally produced foods), opposition (opposition to long-distance food systems and anti-corporativism), and communalization (support of local food, local producers, manufactures, local communities and rural development), using a 7-point interval scale from 1 (strongly disagree) to 7 (strongly agree). The fourth block measured the importance of food attributes and asked participants to select the three most important and the three least important attributes for each category (e.g., “Organic” or “Selected Quality”). The following 10 attributes were considered for determining importance: locally produced, country of origin, region of origin, traditional recipes/production methods, seasonal food, environmentally friendly production, animal welfare, strict quality control, produced on the farm, and traceability of primary ingredients [24]. In the fifth block, the intention to buy local food was measured by two questions: 1) “How likely are you to buy local food?” and 2) “How likely are you to primarily choose a grocery store that offers locally produced food?” A 7-point interval scale ranging from 1 (not at all likely) to 7 (very likely) was used [17]. The last block addressed overall trust in food-supply-chain actors, from local farmers, producers, retailers to national and EU authorities. Trust was measured using a 7-point scale ranging from 1 (very low trust) to 7 (very high trust) [20] (see Appendix A).

### 2.2. Qualitative and Quantitative Data Analysis

Data from the focus groups and shopping with consumers was analyzed using thematic analysis, the most common analysis for qualitative data used to find common patterns in a database [25]. The coding process was conducted to find the key themes. Causal patterns and relationships between and within themes were identified to reveal similarities, differences, and contradictions. Then, an additional thematic analysis of data from shopping with consumers was conducted. The results of the focus groups and shopping with consumers were compared, and the overall results were synthesized into a coherent narrative that included quotes from the analyzed transcript. The analysis was conducted by two experienced researchers. The results of the qualitative study were used to better frame the quantitative study and explain or further illuminate the results of the quantitative analysis. 

The responses to the questionnaire were analyzed using the Statistical Package for Social Sciences version 27.0 (IBM SPSS Statistics), and STATA version 17.0 (StataCorp LLC). Prism version 9.1.0 (GraphPad Software, LLC) was used for developing graphs. Before proceeding with the analysis, the data was screened for missing values, outliers, and normality. Further, they were tested against all assumptions of the statistical methods used. To investigate the importance respondents assign to a list of initiatives/barriers in their purchase decisions for the “Organic” and national “Selected Quality” label, we used the Borda count positional aggregation method [26]. The method assigns a different number of points according to the position of the item in the ranking list of each respondent. The first ranked item receives 10 points, while 6 points are given to the fifth on the list. The points are further summed on a group level for both quality systems. 

Based on the respondents’ choice of trust as top important determinant for food purchases under the analyzed quality systems (“Organic” and “Selected Quality”), two consumer segments were developed, namely “I trust the Organic label” and “I trust the Selected Quality label”. To validate this classification, a separate question was asked, investigating how much respondents’ trust to quality labels, measured using a 7-point Likert-type scale. A mean comparison analysis of both groups against the whole sample using the two-samples independent *t*-test was employed to validate the participants’ group classification.

Descriptive statistics were used to describe the sample and segments regarding their sociodemographic characteristics. The one-sided test of equality for column proportions and the one- and two-sided *t*-test for equality of means were employed to detect the differences between groups. Additionally, multiple correspondence analysis was used to identify the position of two quality systems trustors in relationship to the socioeconomic characteristics. Additionally, the importance of the different food attributes for the identified consumer groups was analyzed. To create an importance scale, the percentage in which an attribute was reported as least important was subtracted from the percentage in which it was reported as most important, resulting in a measure that ranged from 100% to −100% [27,28].

Exploratory factor analysis (EFA) with orthogonal varimax rotation was conducted to determine the principal factors underlying respondents’ understanding, perception, trust, willingness to purchase, and interest in information cues denoting quality, traceability, sustainability, and origin. The latent variables scale internal consistency was measured using Cronbach’s alpha and is presented in the Appendix A. 

Multivariate analysis of variance (MANOVA) was conducted to determine the variability between the groups with respect to the latent variables’ measures. A subsequent *t*-test was conducted to determine significant differences between the groups’ factor scores. Unless otherwise noted, any *p*-value < 0.05 was considered statistically significant.

## 3. Results

Table 2 presents levels of agreements with various statements related to the food labeling schemes (QACS) existing in Slovenia. The result of the analysis shows the highest mean score for agreement with the statement suggesting an existing relationship between labeling schemes and price premiums for foods that have received these labels. Additionally, we observed the lowest variability within this score, which explains the highest agreement in respondents’ answers about the additional value these labels bring to the labeled foods. A high mean score was also observed in the statement concerning trust and in the one describing the higher quality of products under the QACS.

Table 3 presents the results of the analysis of the aggregated ranking scores that the respondents assign to the selected initiatives/barriers in making decisions about certified organic foods, or foods under the national quality scheme “Selected Quality”, respectively. The results show very high agreement in the ranking scores presented by both groups, although for the first ranking item in the initiatives/barriers list, a significant difference in the proportions between the two groups was observed. The top three determinants for purchase are “favorable price to quality ratio”, followed by “trust in the quality scheme” and “product taste”, while the last three are “simplified decision to buy”, “promotion campaign” and “visual appearance”.

Based on the first-ranking selection of the variable “Trust in the quality scheme”, we divided the respondents into two groups, namely “I trust the Organic label” and “I trust the Selected Quality label”. The segment size of the two trustor groups was comparable, with the “I trust the Organic label” group representing 9.63% and the “I trust the Selected Quality label” group representing 10.43% of the total analyzed sample. The result of the validation analysis using the statement “I trust quality labels” of Table 2 proved a significant difference in the mean level of trust between the respondents classified in the group “I trust the Selected Quality label” and the whole sample, which exhibited a lower level of trust in the “Selected Quality” label (mean diff. = 0.35, *CI* = 0.08 − 0.62, *p* = 0.0057). The same was observed for the group “I trust the Organic label”, in comparison with the whole sample, which showed a lower level of trust in the “Organic” label (mean diff. = 0.23, *CI* = −0.03 − 0.49; *p* = 0.042). 

Table 4 presents the descriptive statistics of the sociodemographic data for the entire sample, i.e., the two segments. The descriptive results show some differences in the overall sociodemographic profile of members of the “I trust the Organic label” and “I trust the Selected Quality label” groups in comparison to the general population. The mean age of the two groups is comparable, while notably higher for “Organic” label trustors and significantly higher for “Selected Quality” label trustors compared to the sample of the general population. In terms of income status, a clear difference is observed in the trend of higher income of the group “I trust the Organic label” compared to the group “I trust the Selected Quality label” and the general population sample. The largest proportion of retired consumers is likely to be found in the “I trust the Selected Quality label” group. The focus group analysis revealed that retired consumers place greater trust in organic farming and the “Selected Quality” national label than younger participants and have more positive attitudes towards local foods. This is because they associate them with familiarity, even a nostalgic notion of the good taste of foods from their childhood, or as the 64-year-old retired participant said, 

“I buy meat from a local farmer I know and salami from producers with the “Selected Quality” label because they taste almost the same as when I was a kid. Local food is good and healthy, everything else is artificial products”.

The position closeness of the two segments in relationship to their sociodemographic characteristics is additionally presented in the Appendix A using two dimensions of multiple correspondence analysis.

Notable differences were also found between the two segments in terms of occupational status, with the greater proportion of employed people likely to be in the “I trust the Organic label” group, while students were more likely to be in the “I trust the Selected Quality label” group. Focus group analysis revealed a more complex picture, with students aware of the importance of quality labels and organic foods, but viewing them as too expensive, not justifying that the specific nutritional values are important to them. A typical statement was made by a 20-year-old student, who said, “I know that these foods are good, but they are absolutely too expensive. Also, I am most interested in foods that have more protein and less carbohydrates. I cannot find these qualities in foods with these labels”.

The notable differences between the two groups in terms of education were found among consumers with secondary education, with the “I trust the Organic label” group likely to be more educated, while both groups are more educated compared to the general sample. This has been confirmed by the focus group analysis, which revealed that participants who had completed the secondary school curriculum, which included the topic of food labeling, had more trust in and more positive attitudes towards foods under QACS: 

“Yes, we learned about food labeling in school and that local foods are better because they are healthier and more environmentally friendly” (Participant 1, 23 years old, high school, salesperson).

Figure 1 shows the results of the fourth block of quantitative measures in the survey, in which participants were asked to rate the importance of various food attributes. Figure 1 shows the difference in the assessment for the two consumer segments. The largest difference in attributes influencing food purchases is assigned to the importance of product quality control, which is the most important to the “I trust the Organic label” group and the general population, and less important to the “I trust the Selected Quality label” group. This was also confirmed by the statements in the focus groups, which indicated that organic food consumers are very afraid of fraud and demand constant monitoring to justify the higher price of organic food. “I expect constant monitoring to get what I expect for the price I pay for organic food, without any manipulation,” said the 67-year-old retired physician. Moreover, for the group “I trust the Selected Quality label”, the attribute of local production is the most important, while for the group “I trust the Organic label” it is of secondary importance. It is somewhat surprising that the attribute “environmentally friendly production” is in the fourth place for the group “I trust the Organic label”, but in the third place for the group “I trust the Selected Quality label”. For the general population, we observed the highest score among all for this attribute. Additionally, above the indifference score, we also observed the attributes “farm produced” and “ingredients origin traceability” with the former being more important to the general population, and the latter to trustors of the “Organic” label and the general population. The lowest importance was given to traditional recipes/production methods, and only a slightly higher importance was given to the seasonal food factor. Shopping with consumers revealed that seasonality can influence food purchases without other attributes playing a role. Participants indicated that they were aware of the importance of various attributes, but admitted that they did not consider them for various reasons: 

“I know last time I said that it matters to me if it’s an animal and environmentally friendly product. Well, it’s really important to me, but I rarely have time to look at it closely. I make sure it’s seasonal, and I do not buy strawberries in February because they are artificial,” the 49-year-old employee said.

Country of origin was also rated rather low among the 10 attributes, although the results of the focus group discussions indicated that it was highly important: “I get very annoyed when I buy Chianti wine that is supposedly from Florence but is actually made in China or elsewhere. That’s not right. That’s fraud,” said the 35-year-old lawyer.

A closer look at the product categories (Appendix A) shows that participants buy the largest share of Slovenian food in the milk and dairy products segment (77.2%) and the smallest share in the fruit (55.8%) and vegetable (56.8%) segments. The largest share of local food in Slovenian food is in the meat segment (40.1%) and the smallest in the fruit segment (32.1%). For consumers, locally produced food is most important in the cereals and cereal products, and vegetables and fruit products segments, and quality control in the meat and meat products and milk and dairy products category.

Table 5 presents the results of the MANOVA analysis investigating respondents’ attitudes and perception towards different subjects related to local food (the dependent variables) among three different populations (the nationally representative sample, “I trust the Organic label”, and “I trust the Selected Quality label” groups). Additionally, the results of the mean comparison analysis of latent variables and corresponding dimensions factor scores between the groups are presented. Apart from the latent variable measuring “willingness to purchase local food”, none of the other latent variables or corresponding dimensions showed a significant difference in the variability between the groups. The highest variability within the “willingness to purchase local food” was observed in the general population, while, as expected, the lowest variability was observed in the trustors of local food. Compared to the other two groups, the group “I trust the Organic label” shows notably higher importance of the latent variable denoting interest in quality, traceability, sustainability, and origin of local products. The results also show different attitudes among the groups towards local food with similar behavior in the groups with greater trust in “Organic” and “Selected quality” labels. We can also see significantly and notably different perceptions compared to the general population, specifically in relation to the taste and nutritional composition of local food, as well as to the support of the local food industry, sustainable farming practices, the local community, and rural development. This apparent contrast between groups can be explained by analyzing the statements of focus group participants, which indicated that organic food consumers are more likely to support global rural development that addresses not only environmental but also economic and social challenges. For example, the 33-year-old nurse who trusts the Organic label and buys organic food said: 

“Yes, I trust this label. And I buy organic products, and I want more and more people to buy such products. This is the only right direction for rural development in the less developed parts of the world, you know. It will bring more money, which means better infrastructure development, from schools and roads to digitization around the world. And people will stay at home and not crowd into cities, which is a general trend.”

Respondents in the group “I trust the Selected Quality label” are more likely to support local farmers and producers compared with the general population and show notably higher support for these actors compared with the group “I trust the Organic label”. Additionally, we observed neutral trust with no significant difference between the groups of the manufacturers and retailers that indicate production of local food and in government institutions that monitor compliance with the local production standards. In terms of the understanding of the definition of local food, an interesting trend can be seen showing a higher regional interpretation of the definition of local food in the group “I trust the Selected Quality label” compared with the group “I trust the Organic label” (Appendix A). In one attitude, this is also reflected in the analysis of the focus groups, which reveals a problem with the definition of geographic proximity (local, regional, and national) and the origin/ownership of production and producers. A typical statement comes from a 48-year-old university professor of biotechnology: 

“I do not know what is local here, in Slovenia, because we are so small. When I buy Slovenian cheese or Slovenian apples at the market, it is local for me, even if they were produced on the other side of Slovenia. On the other hand ... I do not know... I am really confused. ... It gets even more complicated with food companies. It’s unclear to me what local food businesses are. Do you mean ownership of a business or food ingredients? For me, Fructal is no longer a Slovenian company, but a Serbian one, and that’s why I do not buy their products anymore, because the money does not stay here. I am really confused.”

Although neutral, a comparably more favorable attitude is observed from the general population towards multinational corporations and large retail chains, and less favorable attitudes towards the origin of products, while the other two groups are more likely to reject food produced by a multinational company and feel uncomfortable eating food with which they are not familiar. This was also confirmed by the focus group analysis. A typical statement from a 52-year-old Master of Pharmacy was: 

“If it’s possible, I want to know where the product comes from, that it’s not made by a multinational company that just manipulates food.” 

There are also differences between consumer groups that show a different willingness to buy local food. Both the “I trust the Organic label” group and the “I trust the Selected Quality label” group are significantly more willing to buy local food and look for a shop that offers local food than the general population. The focus group analysis also confirmed this result in contrast to the analysis of shopping with consumers, as all the focus group participants who stated they would buy local food did not implement this during the shopping observation, mainly because of the high price and taste. A typical statement comes from a 29-year-old fitness instructor: 

“Yes, I know I said that, but I have to admit that I only buy local food when it is cheaper, that is, when the price of these products is generally like those of other products. In that case, of course, I always buy a local product.”

## 4. Discussion

Food consumer research literature has not yet sufficiently tackled the question of similarities and differences between attitudes and expectations of organic food consumers and other consumers. The question is particularly relevant when planning marketing strategies and accompanying support activities. In this respect, it is important to understand whether consumers of organic food and consumers who rely on other QACS in the agri-food system belong to complementary or competitive market segments and how they differ from the general population. This study sought to fill this research gap by using a mixed methods approach with qualitative and quantitative analyses of food consumer preferences and behavior in Slovenia.

The original contribution of this study is to provide further insight about the similarities and differences in the attributes affecting food purchase between the consumers of organic food and consumers who rely on other QACS. Our results show that the “Organic” and the national voluntary “Selected Quality” consumer groups have a similar understanding of the definition of local food, while the regional-based interpretation exceeds the country-based interpretation. However, consumers of organic food are more cosmopolitan and more likely to support the local community, regardless of geographic proximity, than consumers of food with the national “Selected Quality” label. 

In contrast to the study by Ditlevsen et al. [4], which did not find differences between consumers of local, organic, and conventional foods with respect to age, this study showed that consumers who have a very high level of trust in the “Organic” label and in the national voluntary “Selected Quality” label are older than in the general population control sample. Although younger consumers have more knowledge of the food labeling system, since the topic of EU food quality policy is now included in the national secondary school curriculum, they show less trust. This could be explained by their lack of confidence in social institutions [29]. In contrast to Ditlevsen et al. [4], who found that organic consumers are significantly more educated than consumers of local and conventional foods, our study highlighted a statistically significant difference only in the secondary education level for organic trustors, with the two segments likely to be more educated compared with the general population control sample. The study also shows that the largest proportion of retired consumers is found in the “I trust the Selected Quality label” group, and the largest proportion of employed consumers in “I trust the Organic label” group, which could be explained, at least in part, by the more stable incomes of older consumers and the higher incomes of employed consumers. Compared to the segment of consumers with very high trust in organic food labeling, a significantly higher number of college students are likely to be found in the general population, primarily due to the higher price of organic food, according to the focus group analysis. The high cost of food, especially organic and local food with the QACS label, is the biggest barrier for consumers to buying such products [29,30,31]. In contrast to consumers in Western Europe, who prioritize taste [32], price or a trade-off between price and quality is an important purchasing factor for consumers in Central and Eastern Europe [31,33]. Part of this behavior can be attributed to purchasing power, and part to personal values. Compared to the inhabitants of other Central and Eastern European countries, Slovenians spend a large part of their financial resources on transport and a small part on food [34], which means that they prefer channeling their disposable income into luxury or status items over food quality. 

The results of the analysis of the underlying dimensions of the latent variables are somewhat consistent with what the literature has shown [15,16,17,23]. 

Regarding the importance of attributes, the correspondence analysis identified two dimensions that effectively distinguish the two segments and proved the suitability of trust in a quality system as a segmentation variable that can correctly classify consumers, especially those who have very high trust in organic food labeling and have proven to be those who value local food, as described in the literature [4]. 

The study found that both consumers with very high trust in organic food labeling and those trusting the “Selected Quality” national scheme were more likely to purchase local foods and search for a shop that offered local foods than the consumer part of the general population. However, the analysis of shopping with consumers does not confirm this. Thus, this discrepancy suggests that declarative statements do not (always) find confirmation in behavior and that the use of different methods is useful to verify results based on declarative statements. The so-called attitude/value-behavior gap [35,36], which occurs when positive attitudes do not translate into consistent purchase intentions or actual purchase behavior, is well known in the literature and one of the biggest challenges for researchers, the food industry, and marketers because different consumer segments are influenced by different factors, the most problematic of which are automatic and intuitive decision-making processes, such as habits [37]. 

A vague understanding of the definition of local food regardless of consumer segments, as pointed out by many authors [15,16,17], was particularly evident in the focus group analysis, as participants questioned what “local” means in a small country like Slovenia and revealed an arbitrary selection of local foods. This also shows that attitudes towards local food are based on the affective component rather than the cognitive component. The study found that there are differences between the consumers who have very high trust in the “Organic” label and the group that has very high trust in the “Selected Quality” label when it comes to supporting local farmers and producers. Thus, the “Organic” consumer group is more likely to support the local community, regardless of geographic proximity (i.e., global), than the “Selected Quality” consumer group. Therefore, our results are consistent with findings from other studies [4,5,6] that organic food consumers have a less strict understanding of locality, while proponents of “local food” are more likely to support proximity and, in particular, local farmers and producers in their immediate vicinity. In contrast to a previous study [4,5,6], the results of our quantitative study show that consumers with very high trust in organic food include environmental considerations rather low in their list of most important attributes when purchasing food, which is similar in comparison with the consumers with high trust in the national “Selected Quality” label. 

Therefore, this study showed that there is a need to pay much more attention to consumers’ overall understanding of local food and their trust in local food actors, farmers, businesses, and organizations. Consumer interest and search for transparent information about food quality, traceability, origin, and sustainability should not be neglected in supporting demand for foods with quality labels and QACS. It is not appropriate to focus a communication or promotion campaign on attributes that appeal primarily to general consumers, such as seasonal ingredients, on-farm production, country of origin, animal welfare, and traditional recipes. On the contrary, the consumers with high trust in the “Organic” label would react to attributes such as: strict quality control, traceability, and region of origin; the same holds true for the consumers with high trust in the national voluntary label “Selected Quality”. Consumer education, constant and systematic promotion and communication of the QACS, and providing support to improve distribution on the markets, are needed to strengthen the economic development of the niche sector.

Although the study provided valuable insights into the niche segments of the food market, this analysis is limited by the fact that it was conducted during the COVID-19 pandemic, when various restrictive measures were imposed in response to the pandemic. This also explains why focus groups and the quantitative survey were fully conducted on-line. Previous reports have highlighted that the epidemic affected consumer food-related behaviors, but it is not clear whether such changes are permanent [38]. We also suggest that future studies look deeper into the differences in perceptions and resulting consumption patterns between different food categories, as our study has shown different purchasing patterns with regard to national and locally produced food. 

## 5. Conclusions

The combined methods approach using shopping with consumers, qualitative focus groups, and a quantitative survey provide data that shows a very high level of agreement in the results obtained, suggesting that there are different segments of the population willing to buy local food, although they have different opinions regarding the financial value of such food. The “Organic” and national voluntary “Selected Quality” consumer groups have similar understandings of local food, with region-based interpretations outperforming country-based interpretations. The “Organic” group was more cosmopolitan and supportive of the local community, regardless of geographic proximity, than the “Selected Quality” group. Older consumers occupy a larger share of both segments, with professionals and individuals with higher incomes more likely to be in the “Organic” group and retirees and students more likely to be in the “Selected Quality” group. To increase the consumers’ interest in food with the “Organic” and “Selected Quality” schemes, more specific product propositions should be developed.

## Figures and Tables

**Figure 1 foods-12-01132-f001:**
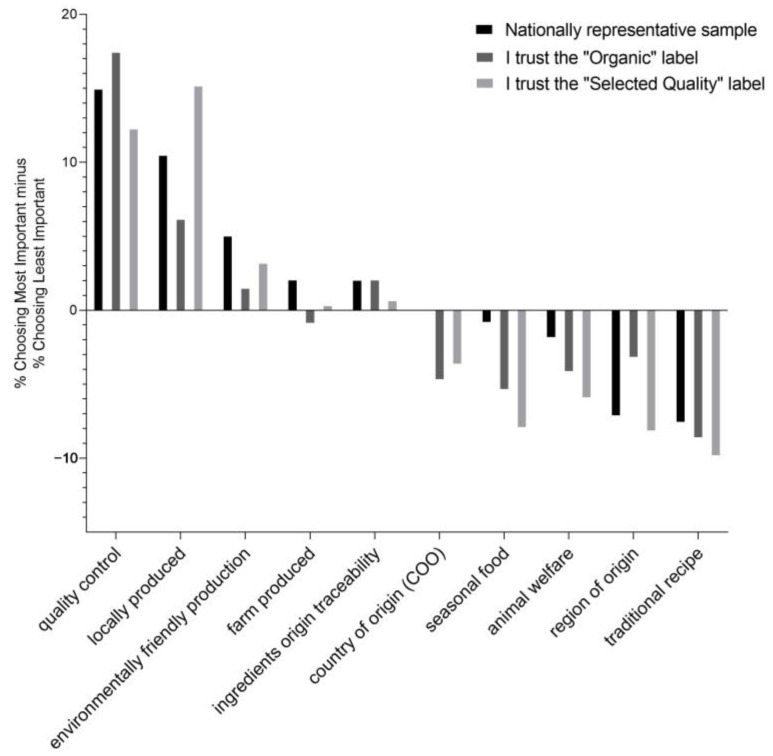
Differences in Perceived Importance of Food Attributes (Note. Attribute relative importance, ranked by the nationally representative sample).

**Table 1 foods-12-01132-t001:** Sociodemographic Characteristics of the Study Sample (*n* = 1007).

Variables	Levels	*n*
Gender	Male	504
	Female	503
Age (years)	Mean (SD)	49.4 (16.3)
Age groups (years)	18–24	83
	25–34	144
	35–44	182
	45–54	175
	55–65	172
	65+	251
Education	Elementary school and less	21
	High school	576
	College and university degree	359
	Postgraduate degree	51
Income	Much lower than the average salary	281

**Table 2 foods-12-01132-t002:** Levels of Agreement with Statements Related to Food Labeling Schemes (*n* = 1007).

Statements	Mean (SD)
The label makes the food more expensive.	5.34 (1.48)
The label on the food guarantees higher quality.	4.72 (1.58)
I trust quality labels.	4.61 (1.54)
When buying food, I usually prefer to buy food with a quality label.	4.37 (1.66)
I am willing to pay more for food with a quality label.	4.37 (1.68)
The label makes it easier for me to choose food.	4.28 (1.67)

Note. The agreement with statements was measured on a 7-point Likert-type scale ranging from 1 (strongly disagree) to 7 (strongly agree).

**Table 3 foods-12-01132-t003:** Ranking of Factors Determining the Purchase of Certified Organic Food or Food under the National Quality Scheme “Selected Quality” (*n* = 1007; two subsamples).

List of Initiatives/Barriers	Organic Label *n* (%)	Borda Count	Rank	Selected Quality Label *n* (%)	Borda Count	Rank
Label familiarity	31 (6.4)	1740	6	38 (7.3)	1713	6
Favorable price-to-quality ratio	144 (29.5) ^a^	3257	1	127 (24.5) ^b^	3480	1
Taste	73 (15.0)	2692	3	87 (16.8)	2901	3
Visual appearance	16 (3.3)	1342	8	20 (3.9)	1323	8
Availability	37 (7.6)	2150	4	36 (6.9)	2454	4
Wide product range	18 (3.7)	1546	7	27 (5.2)	1527	7
Institutional trust	49 (10.0)	2055	5	42 (8.1)	2063	5
Trust in the quality scheme	96 (19.7)	2853	2	105 (20.2)	3209	2
Simplified decision to buy	13 (2.7)	565	10	20 (3.9)	731	10
Promotion campaigns	11 (2.3)	863	9	17 (3.3)	1049	9
Total *n* (%)	488 (100)			519 (100)		

Note. Values in the same row and sub-table sharing different superscript letters are significantly different at *p* < 0.05 in the one-sided test of equality for column proportions. The test assumes equal variances.

**Table 4 foods-12-01132-t004:** Sociodemographic Characteristics of the Total Sample and Segments (*n* = 1007).

Variables	Nationally Representative Sample *n* (%)	“I Trust the Organic Label” *n* (%)	“I Trust the Selected Quality Label” *n* (%)
**All**	1007 (100)	96 (100)	105 (100)
**Age Mean (SD)**	52.1 (15.8) ^#^	54.7 (15.3)	56.6 (15.3) ^#^
Gender			
Male	521 (51.7)	53 (55.2)	55 (52.4)
Female	486 (48.3)	43 (44.8)	50 (47.6)
**Region**			
Eastern Slovenia	497 (52)	43 (50.0)	51 (50.5)
Western Slovenia	458 (48)	43 (50.0)	50 (49.5)
**Education**			
Lower secondary	495 (49.2)	40 (41.7)	51 (48.6)
Secondary	231 (22.9) ^†^	32 (33.3) ^†^	24 (22.9)
Post-secondary and tertiary	281 (27.9)	24 (25.0)	30 (28.6)
**Income**			
No income	35 (4.1)	2 (2.4)	4 (4.5)
Lower than average	462 (54.3) ^†^	36 (42.9) ^†^	45 (51.1)
Average	170 (20.0) ^†^	25 (29.8) ^†,‡^	15 (17.0) ^‡^
Higher than average	184 (21.6)	21 (25.0)	24 (27.3)
**Employment**			
Student	74 (7.5) ^†^	2 (2.1) ^†^	6 (5.7)
Employed	494 (49.9)	51 (54.3) ^‡^	45 (42.9) ^‡^
Unemployed	57 (5.8)	4 (4.3)	4 (3.8)
Retired	365 (36.9)	37 (33.4) ^‡^	50 (47.6) ^‡^

Note. Where not indicated next to the variable’s name, the values in brackets represent percentage as indicated in the column name. Values in the same row sharing the same superscript symbol (#, †, ‡) are significantly different at *p* < 0.05 in the one-sided test of equality for column proportions or one-sided *t*-test for equality of means. The tests assume equal variances.

**Table 5 foods-12-01132-t005:** Results of the Multivariate Analysis for the Hypothesis Testing of Variance Difference and Mean Comparison Analysis.

Latent Variables and Dimensions	Analysis of Variance Testing			Independent Samples *t*-Test		
	Wilk’s Lambda	*F*	*p*-Value	“I Trust the Organic Label” Factor Scores Mean (*SD*)	“I Trust the Selected Quality Label” Factor Scores Mean (*SD*)	General Population Sample Factor Scores Mean (*SD*)
**Interest in information cues denoting quality, traceability, sustainability, and origin** (5 items)	0.994	0.70	N.S.	0.10 (0.94)	0.02 (0.91)	−0.01 (0.90)
**Understanding definition of local food**						
*Regional-based interpretation* *(3 items)*	0.996	0.78	N.S.	0.02 (0.74)	0.08 (0.73)	−0.01 (0.73)
*Country-based interpretation* *(2 items)*	0.998	0.47	N.S.	0.002 (0.63)	−0.03 (0.63)	0.003 (0.65)
**Attitudes towards local food**						
*Lionization and opposition* *(6 items)*	0.988	1.23	N.S.	−0.02 (0.93)	0.15 (0.75) #	−0.01 (0.91) #
*Communalization* *(3 items)*	0.992	1.63	N.S.	0.11 (0.74)	0.15 (0.84)	−0.03 (0.82)
**Trust in actors producing local food**						
*Person-related dimension* *(2 items)*	0.997	0.96	N.S.	−0.01 (0.90)	0.17 (0.86) #	−0.02 (0.93) #
*Institution-/Corporation-related dimension* *(3 items)*	0.999	0.25	N.S.	0.05 (0.94)	−0.05 (0.89)	0.00 (0.90)
**Willingness to purchase local food** (2 items)	0.988	3.61	0.06	0.21 (0.85) †	0.23 (0.77) #	−0.04 (0.94) †,#

Note. The table shows the mean comparison analysis of the factor scores of: (a) general population sample; (b) “I trust the Organic label”, “I trust the Selected Quality label”. N.S. denoting a result from a statistical hypothesis-testing meaning not statistically significant. Values in the same row sharing the same superscript symbol (#, †) are significantly different at *p* < 0.05 in the two-sided *t*-test for equality of means. The tests assumes equal variances.

## Data Availability

The data presented in this study are available on request from the corresponding author.

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
