# Peer review of "Identifying Differences in Consumer Attitudes towards Local Foods in Organic and National Voluntary Quality Certification Schemes"

_foods, 2023, doi:10.3390/foods12061132_

Round 1

Reviewer 1 Report

Well-written manuscript. Very smooth introduction, discussion, and conclusions.

However, some data are not presented clearly, leaving room for ambiguity or misleading the reader. Please fix these issues (detailed in the file)

Author Response

Response to Reviewer 1 Comments

Thank you for the very useful comments. All the suggested improvements highlighted in the text of the manuscript have been reviewed and the manuscript text has been reworked according to your suggestions. Where felt necessary the corrections have been made, otherwise explanation is provided below.

Reply: In the line 70 the suggestion was accepted and during shortening of the introduction the whole paragraph, where the text related to your comment was included was deleted.

Reply: In the line 377 the sentence is presented in paste tense “Descriptive statistics was used…” as this is usually recommended, when writing the methodology part.

Reply: In the line 382 the use of the multiple correspondence analysis statistical method is indicated. The text in the line 482 of the results section is now changed and instead of “Table S1” is now changed to “Figure S1”. We have additionally clarified in the text by adding: “using two dimensions of multiple correspondence analysis”.

Reply: In the line 414 the suggestion is accepted, and below the table a new note is added: “The agreement with statements was measured on 7-point Likert-type scale ranging from 1 (strongly disagree) to 7 (strongly agree)”.

Reply: In the line 425 we reworked the text to be more understandable of what it meant to say, by adding: “the first ranking item in the initiatives/barriers list”.

Reply: In the line 435 we have corrected the text as suggested by adding “different superscript”.

In the line 456 there was question asking: “Where do these data come from? From table 3?“

Reply: Thank you very much for your question. The data comes from analysis of a question which is presented in the Table 2 under the statement “I trust quality labels”. The procedure of validation of segmentation is explained in the methodology section and additionally clarified in the results section. The paragraph in the line 368 of the methodology section explaining validation was reworked to clarify this:

“Based on the respondents’ choice of trust as top important determinant for food purchases under the analyzed quality systems (“Organic” and “Selected Quality”), two consumer segments were developed, namely “I trust the Organic label” and “I trust the Selected Quality label”. To validate this classification, a separate question was asked, investigating how much respondents’ trust quality labels, measured using a 7-point Likert-type scale. A mean comparison analysis of both groups against the whole sample using the two-samples independent t-test was employed to validate the participants’ group classification.”

In the line 465 there was question regarding table 4, and thus “Are these difference between column proportions significant or not. It is not clear from Table 4.”

Reply: Thank you very much for your question. Our answer is that the values in the same row where the superscript symbols (#, †, ‡) are the same are significantly different at p < 0.05.

In the line 614 there was question asking: “Please explain what these superscripts are?”

Reply: Thank you very much for your comment. In the line 617 we have provided explanation about the symbols.

Reviewer 2 Report

The manuscript "Local food attitudes and behavior: How do consumers compare in organic and national voluntary quality certification schemes" is an interesting topic. However, corrections (attachment) are needed to improve your submission for Foods journal.

General comment:
The subject is interesting, from the point of view of health. At present people are more concerned
about consuming quality food. However, the article is presented in an inconspicuous way and is
too extensive and over-explained. A selection of the information presented is recommended, to
avoid repetition between sections.
The authors did not place the numbering of the lines throughout the manuscript, which makes
revision difficult, since it is not possible to refer to certain lines.
Specific comments:
Title
The title is very subjective, it does not clearly reflect the objectives or results of the study, even
the title seems appropriate for a review article. It is recommended to reformulate the title.
Abstract
The abstract is good however, it is necessary to add a couple of introductory lines explaining the
reason or importance of the study.
Keywords
The keywords must be that words, that allow the article to be found in the search. The authors put
in too many compound words that look like sentences. I suggest that you use a selection of the
most representative ones that will help the search for your article.
1. Introduction
The introduction has very good information, it puts the problem in context and shows related
studies. However, since the article is categorized as an "original article", it is necessary to
synthesize the information to have a maximum of 2 pages of introduction. Prioritizing
contextualizing, addressing the problem and the proposed solution.
2. Materials and methods
Homogenize the formulation of statements, on some occasions they use terms in the first person
in the plural "we took photos" and in other cases "photos were taken"; it is recommended to set
all verbs to infinitive.
The methodologies are overdetailed, the authors make a narrative of what was done and how it
was done, however, it is important to place the information as a "recipe" where the steps and
important data are included.
In the data analysis section, previously detailed information is repeated. The analysis is over
explained.
3. Results
Table 3 and 5. The title and footer of the table must be adjusted to the length of the table.
The authors separated the results and discussion sections, however, in the results section they
show us discussions or assertions made with an analysis of the data and that is already discussion.
It is recommended to reconsider the union of both sections (results and discussion) or totally
separate the assertions from the results.
4. Discussion
This section is adequate; however, it repeats part of the results previously mentioned in section
3.
5. Conclusions
The conclusions take up so much information from the study that it seems like a discussion
section. The conclusions must be concise, and representative of the results and analysis carried
out previously. The statements should not be longer than half a page.
5. References
The number of references is adequate, the years are current. References must be adjusted to the
format of the journal.

Author Response

Response to Reviewer 2 Comments

Thank you for the very useful comments. Please find below our answers.

Comment regarding the title: “The title is very subjective, it does not clearly reflect the objectives or results of the study, even the title seems appropriate for a review article. It is recommended to reformulate the title.”

Reply: Thank you very much for your comments. The title has been reworked and the new title is “Identifying differences in consumer attitudes towards local foods in organic and national voluntary quality certification schemes.”

Comment regarding the abstract: The abstract is good however, it is necessary to add a couple of introductory lines explaining the reason or importance of the study.

Reply: Thank you for your suggestion. On your recommendation, we have added the importance of the study: “As no study on attitudes towards local food has compared "Organic" and national quality scheme consumer segments, the study aimed to provide further insights and clarifications on the issue of consumer segmentation in terms of trust to organic food and food of selected quality perceived as local, along socioeconomic characteristics, and other important determinants of this complex interaction”.

Comment regarding the keywords: “The keywords must be that words, that allow the article to be found in the search. The authors put in too many compound words that look like sentences. I suggest that you use a selection of the most representative ones that will help the search for your article.”

Reply: We corrected as suggested. The new keywords are: quality schemes, Selected Quality, organic label, trust, segmentation, local food, consumer attitudes.

Comment regarding the introduction: “The introduction has very good information, it puts the problem in context and shows related studies. However, since the article is categorized as an "original article", it is necessary to synthesize the information to have a maximum of 2 pages of introduction. Prioritizing contextualizing, addressing the problem and the proposed solution.”

Reply: We greatly appreciate your helpful and constructive comments. Indeed, it was necessary to synthesise the information to produce a maximum 2-page introduction. We have prioritised, contextualised the problem and the proposed solution.

Comment regarding the methodology section: “Homogenize the formulation of statements, on some occasions they use terms in the first person in the plural "we took photos" and in other cases "photos were taken"; it is recommended to set all verbs to infinitive. The methodologies are overdetailed, the authors make a narrative of what was done and how it was done, however, it is important to place the information as a "recipe" where the steps and important data are included. In the data analysis section, previously detailed information is repeated. The analysis is over explained.”

Reply: Thank you for noting this, where possible, we homogenised the formulation of statements to be linguistically correct. We simplify the presentation of information in the methodology section to level not disrupting the presented structure and being precisely informative for the analysis of the important data.

Comment regarding the results section: “Table 3 and 5. The title and footer of the table must be adjusted to the length of the table. The authors separated the results and discussion sections, however, in the results section they show us discussions or assertions made with an analysis of the data and that is already discussion. It is recommended to reconsider the union of both sections (results and discussion) or totally separate the assertions from the results.”

Reply: As suggested, we have separated the assertions from the results.

Comment regarding the discussion section: “This section is adequate; however, it repeats part of the results previously mentioned in section 3.

Reply: As suggested, we have shortened the discussion by deleting repeated results.

Comment regarding the conclusion section: “The conclusions take up so much information from the study that it seems like a discussion section. The conclusions must be concise, and representative of the results and analysis carried out previously. The statements should not be longer than half a page.”

We greatly appreciate your helpful and constructive comment. We have radically shortened the conclusion:

“The combined methods approach using shopping with consumers, qualitative focus groups, and a quantitative survey provide data that shows a very high level of agreement in the results obtained, suggesting that there are different segments of the population willing to buy local food, although they have different opinions regarding the value of such food. The “Organic” and the national voluntary “Selected Quality” consumer groups have similar understandings of local food, with region-based interpretations outperforming country-based interpretations. The “Organic” group was more cosmopolitan and supportive of the local community, regardless of geographic proximity, than the “Selected Quality” group. Older consumers occupy a larger share of both segments, with professionals and individuals with higher incomes more likely to be in the “Organic” group and retirees and students more likely to be in the “Selected Quality” group. To increase the consumers’ interest in food with the “Organic” and “Selected Quality” schemes, more specific product propositions should be developed.”

Comment regarding the references:” The number of references is adequate; the years are current. References must be adjusted to the format of the journal.”

Reply: Thank you for noting this, references were adjusted to the format of journal.

Round 2

Reviewer 1 Report

The manuscript has been improved. Reading is now even smoother. It is accepted in the present form.

Reviewer 2 Report

The changes were successful and the article has been improved.